# Macrophage Profiling in Head and Neck Cancer to Improve Patient Prognosis and Assessment of Cancer Cell–Macrophage Interactions Using Three-Dimensional Coculture Models

**DOI:** 10.3390/ijms241612813

**Published:** 2023-08-15

**Authors:** Nour Mhaidly, Fabrice Journe, Ahmad Najem, Louis Stock, Anne Trelcat, Didier Dequanter, Sven Saussez, Géraldine Descamps

**Affiliations:** 1Department of Human Anatomy and Experimental Oncology, Faculty of Medicine, Research Institute for Health Sciences and Technology, University of Mons, Avenue du Champ de Mars, 8, 7000 Mons, Belgium; nour.mhaidly@umons.ac.be (N.M.); fabrice.journe@umons.ac.be (F.J.); louis.stock@student.umons.ac.be (L.S.); anne.trelcat@umons.ac.be (A.T.); sven.saussez@umons.ac.be (S.S.); 2Laboratory of Clinical and Experimental Oncology (LOCE), Institute Jules Bordet, Université Libre de Bruxelles (ULB), 1000 Brussels, Belgium; ahmad.najem@bordet.be; 3Department of Otolaryngology and Head and Neck Surgery, CHU Saint-Pierre, 1000 Brussels, Belgium; didier.dequanter@pandora.be

**Keywords:** head and neck cancer, cancer cell lines, monocytes, macrophage, polarization, scoring system, 3D coculture, spheroid

## Abstract

Tumor-associated macrophages are key components of the tumor microenvironment and play important roles in the progression of head and neck cancer, leading to the development of effective strategies targeting immune cells in tumors. Our study demonstrated the prognostic potential of a new scoring system (Macroscore) based on the combination of the ratio and the sum of the high and low densities of M1 (CD80+) and M2 (CD163+) macrophages in a series of head and neck cancer patients, including a training population (*n* = 54) and a validation population (*n* = 19). Interestingly, the Macroscore outperformed TNM criteria and p16 status, showing a significant association with poor patient prognosis, and demonstrated significant predictive value for overall survival. Additionally, 3D coculture spheroids were established to analyze the crosstalk between cancer cells and monocytes/macrophages. Our data revealed that cancer cells can induce monocyte differentiation into protumoral M2 macrophages, creating an immunosuppressive microenvironment. This coculture also induced the production of immunosuppressive cytokines, such as IL10 and IL8, known to promote M2 polarization. Finally, we validated the ability of the macrophage subpopulations to induce apoptosis (M1) or support proliferation (M2) of cancer cells. Overall, our research highlights the potential of the Macroscore as a valuable prognostic biomarker to enhance the clinical management of patients and underscores the relevance of a spheroid model in gaining a better understanding of the mechanisms underlying cancer cell–macrophage interactions.

## 1. Introduction

Head and neck cancer (HNC) is the seventh most common type of cancer worldwide, with more than 900,000 new cases and 467,000 deaths in 2020 [1]. The vast majority of HNCs are squamous cell carcinomas, and they are divided into several subtypes according to the primary tumor localization (oral cavity, nasopharynx, oropharynx, hypopharynx, and larynx) [2]. Head and neck tumors are associated with high levels of morbidity and mortality [3]. The overall 5-year survival rate of HNC is less than 50%, and this has remained unchanged over the past several decades [4], although novel therapeutic strategies have been proposed.

HNC is surrounded by a complex environment including blood and lymph vessels, fibroblasts, endothelial cells, and immune cells (producing growth factors and cytokines) within an extracellular matrix [5]. Cross-talks between the cancer cells and the cells from the tumor microenvironment (TME) support tumor development, impact therapeutic responses, and ultimately affect patient survival [6]. In particular, immune cells, including myeloid-derived suppressor cells, macrophages, dendritic cells, lymphocytes, and natural killer cells, are important players in tumor-mediated immune suppression, favoring tumor progression. In this context, several studies showed that HNCs are highly infiltrated by macrophages (up to 30%) and that such cancers presenting higher macrophage infiltrations were correlated to shorter recurrence-free survival (RFS) and overall survival (OS) [7,8,9,10,11]. Within the tumors, macrophages are named tumor-associated macrophages (TAMs) and constitute a link between innate and adaptative immune responses. Macrophages arise from bone marrow-derived monocytes and circulate in the bloodstream before being recruited into tumors by chemoattractants like CCL2/Monocyte chemoattractant protein (MCP-1) or colony-stimulating factor 1 (CSF-1) [12,13].

According to their high phenotype plasticity and depending on cytokine stimuli, TAMs present two main phenotypes, defined as M1 and M2, associated with either anti- or protumoral activities, respectively [14]. M1-polarized macrophages are activated by Th1 cytokines, including interferon-γ (IFNγ), or following the binding of lipopolysaccharide (LPS) by its Toll-like receptors (particularly TLR4), leading to the production of reactive oxygen species (ROS), nitric oxide (NO), and pro-inflammatory cytokines (IL1, IL6, IL12, and TNFα) [15,16]. In contrast, M2-polarized macrophages are activated by Th2 cytokines like IL4 and IL13 and, consequently, produce anti-inflammatory cytokines such as IL10, IL8, and TGFβ as well as chemokines CCL18 and CCL22. Phenotypically, M1 expresses the costimulatory molecules B7.1 (CD80) and B7.2 (CD86), which may interact with CD28 and CD152 (CTLA4) receptors on T cells, while M2 expresses the hemoglobin receptor (CD163) and a cell surface scavenger receptor (CD206) [17].

Recently, M2-TAMs have become relevant therapeutic targets due to their essential roles in tumorigenesis. In fact, they are implicated in angiogenesis, migration and invasion, epithelial-to-mesenchymal transition (EMT), intravasation and extravasation, and, finally, immunosuppression through PD-L1/PD-L2 expression, IL-10, TGF-β, arginase-1, and prostaglandins production [18,19]. Moreover, their interactions with cancer cells may modify their phenotypes and functions. Therefore, a thorough understanding of such interactions is very important to unveil the mechanisms underlying tumor development in the face of M2-TAMs.

A three-dimensional (3D) model may be helpful to create a tissue architecture that can mimic the TME. Indeed, the 3D culture system allows cell–cell and cell–matrix interactions, recapitulating in vivo tumor cell characteristics including growth kinetics, nutrient gradients, gene expression, and hypoxic tumor regions [20,21,22,23]. The spheroid aggregation of cells is one of the easiest ways to culture tumor cells in three dimensions [21]. The self-aggregation of cells, the reproducibility of the formation, and the structural similarities with native tissues make the spheroid a useful tool for the culture of “mini-tumors” [24,25,26]. Regarding their morphology, the spheroid is divided into three layers: a peripheral layer of proliferating cells, an intermediate one of quiescent cells, and a necrotic layer of cells caused by hypoxia. This complex structure creates a physiochemical gradient of O_2_ as well as a decrease in the nutrient gradient with the depth of the spheroid. On the other hand, the accumulation of metabolic waste is higher in the core of the spheroid than in the peripheral layer [27,28], also mimicking in vivo tumors.

This study aimed to analyze the macrophage profile (including phenotype and cell recruitment) in 54 tumors from HNC patients (training population), with the goal of establishing a Macroscore that correlates with patient survival. The clinical value of the Macroscore was then validated using a series of 19 new tumors from HNC patients (validation population). Additionally, we developed a 3D coculture model using HNC cell lines and monocytes to investigate how cancer cells control monocyte infiltration into the spheroid and thereby regulate their phenotype and growth. Such a model will help to better understand tumor–macrophage interactions and mechanisms impacting tumor progression [29,30].

## 2. Results

### 2.1. Macrophage Scores and Patient Survival (Training Analysis)

We explored the distinct macrophage subpopulations present in a population of 54 patients with HNC (Table 1) by evaluating M1 macrophage marker CD80 and M2 macrophage marker CD163. The results of our CD163 and CD80 immunostaining are illustrated in Figure 1. The results show different densities of macrophage infiltration (M1/CD80 or M2/CD163) in both the tumor stroma (ST) and intratumoral (IT) areas. Based on our quantitative analysis of ST and IT compartments and by calculating total (tot) macrophages as the sum of both compartments (ST + IT), we used two cut-offs: 2.9 for the ratio of M2_tot_/M1_tot_ and 100 cells/mm^2^ for the quantity of total macrophages (M1_tot_ + M2_tot_) counted by mm^2^. The photos depict tumors with high infiltration of CD163+ macrophages (1A) versus low infiltration of CD163+ macrophages (1B), and we observed that they are predominantly located in ST. Figure 1C, D also demonstrate a high number of CD80+ macrophages in some patients and low density in others.

Based on the ratio of M2_tot_/M1_tot_ or the sum M1 + M2 of macrophages, we established groups of patients with a high M2/M1 ratio or a high M1 + M2 density and evaluated their prognostic values regarding patient OS. Univariate Cox regression analysis revealed that both scores were significantly associated with OS (*p* = 0.02). Indeed, Figure 2A shows that a higher ratio of M2/M1 macrophages was associated with a poorer prognosis for HNC patients. We also observed that a higher value of M2 + M1 macrophages was associated with shorter survival (Figure 2B). Interestingly, these two variables were independent regarding a multivariate analysis (Table 2). These two macrophage scores being significant as prognostic factors with high HR and low *p*-values (*p* < 0.005), we combined these two scores into one, called the “Macroscore”, and evaluated its prognostic performance. The results showed that HNC patients with a high Macroscore had significantly poorer survival than those with a low score (*p* = 0.002, Figure 2C), meaning that patients with either a high ratio of M2/M1 and/or a high value of M1 + M2 (based on previous cut-offs) have shorter survival.

Finally, Cox multivariate analyses were performed including either p16 status (Table 3) or tumor stage (Table 4) to identify the independent contribution of our “macrophage” scores regarding OS. Our results showed that both scores, based on either macrophage quantity or phenotype, were independent prognostic factors compared to the conventional clinical parameters. Furthermore, the Macroscore that we calculated was found to be a better predictor of patient survival (*p* = 0.006) than the p16 status and the patient tumor stage (Table 5).

### 2.2. Validation of the Macroscore

To assess the reliability of the Macroscore obtained from the training population (Table 6), an independent validation population consisting of 19 HNC patients (Table 6) was used. The analysis revealed a significant association between high Macroscore values and shorter overall survival (OS) in comparison to patients with low Macroscore values (Cox regression, *p* = 0.017) (Figure 3).

To further evaluate the predictive capability of the Macroscore, cross-tabulation analyses were performed on the validation population, allowing us to determine sensitivity, specificity, and positive/negative predictive values (PPV and NPV) based on high/low Macroscore versus patients who were deceased or alive at 34 months. The results demonstrated that the Macroscore exhibited excellent predictive performance, with a good specificity of 78% and sensitivity of 90%. Moreover, it showed good PPV of 82% and NPV of 88% (Chi-square test, *p* = 0.003). Therefore, evaluating the Macroscore could provide complementary information, refining the conventional diagnostic markers.

### 2.3. Formation and Characterization of Spheroids

We investigated the ability of our cancer cell line to form 3D spheroids in order to mimic the in vivo environment. Regarding the monoculture of the FaDu cancer line, cells were cultured in nonadherent 96-U-well plates. They started to aggregate on day 2 and formed spheroids after 3 days, demonstrating a compact structure without a necrotic core. The proliferation was stopped when spheroids reached a diameter between 400 and 500 µm to avoid the occurrence of an excessive necrotic core (Appendix A). For coculture experiments, spheroids were composed of cancer cells and human monocytes. The ratio was optimized to represent the in vivo situation, and our results showed that spheroids plated with four times more monocytes than cancer cells were composed of the optimal ratio to illustrate the infiltration of immune cells in the tumor (about 10 to 20% of infiltrating macrophages at the end of experiments). It is important to note that once monocytes were isolated, they polarized in contact with cancer cells and poorly proliferated, unlike cancer cells.

### 2.4. Impact of Macrophage Subpopulations on Cancer Cells Development

We then assessed the impact of FaDu tumor cells on the polarization state of monocytes CD14+ in the coculture spheroid model. Four types of spheroids were compared: spheroids containing FaDu cells alone, FaDu cells + monocytes, FaDu cells + M1 generated by the addition of pro-inflammatory activators (LPS and IFNγ), and FaDu cells + M2 induced by the addition of anti-inflammatory cytokines (IL4 and IL13). In each condition, eight spheroids were photographed on different days of their development (days 2, 5, and 7) to compare the growth evolution in each coculture (Figure 4A).

We observed that the difference between our four conditions became significant on day 5. Indeed, the development of FaDu + monocytes was significantly larger with a volume of 0.02 mm^3^ compared to FaDu alone, measuring 0.015 mm^3^, suggesting that the addition of monocytes promoted the proliferation of cancer cells (*p* = 0.04) (Figure 4B). Compared to FaDu + monocytes, a slowing down of the growth of FaDu + M1 can be observed from day 5, and such a difference became significant on day 7, where the volume of FaDu + M1 stabilized at 0.025 mm^3^ (*p* = 0.02), while the development of FaDu + M2 grew significantly to 0.055 mm^3^ (*p* < 0.001). Moreover, we observed that the volume of the FaDu + M2 spheroids was twofold higher than that of FaDu alone (*p* < 0.001). In addition, we showed a significant difference between the volume of FaDu + M1 (0.025 mm^3^) and FaDu + M2 (0.055 mm^3^) on day 7 (*p* < 0.001). These results agree with the demonstrated functions of M1 and M2, known to be, respectively, anti- and protumoral macrophages, and emphasize the differential effects of the two distinct subpopulations of macrophages in our 3D model.

As shown in Figure 4C, D, the percentage of proliferative cells evaluated with the proliferation marker Ki-67 in FaDu spheroids rose to 80%. This percentage showed a significant decrease in M1–FaDu spheroids with only 20% of Ki-67-positive cells (*p* < 0.001), while it was 60% in M2–FaDu spheroids (*p* = 0.04). Hence, the comparison of M1 and M2 coculture conditions showed a significant difference in Ki-67-positive cells, supporting the idea that M1 inhibits the development of the cocultured spheroids (*p* = 0.03) (Figure 4D). Of note, cocultures of cancer cells with monocytes did not significantly affect the growth of the spheroids, as compared to the FaDu-alone condition.

### 2.5. Influence of Distinct Spheroid Conditions on the Apoptotic Profile

Having observed that FaDu cells continue to form larger spheroids in the presence of monocytes and M2, while M1 macrophages inhibited spheroid formations, we were interested in the apoptotic profiles of the different coculture conditions (Figure 5). To investigate this, on day 7 of coculture, spheroids were dissociated and labeled with the Annexin V/dead cell marker. As presented in Figure 5, M1–FaDu generated the highest percentage of apoptosis with more than 60% of apoptotic cells. This rate was significantly higher than those observed in the other coculture conditions except for M2–FaDu. Altogether, we reported no significant changes in the apoptotic profile between FaDu, monocytes–FaDu, and M2–FaDu with, respectively, 33.43, 35.05, and 40.77% of apoptotic cells. These data suggest that M1 may participate in tumor damage by triggering the initiation of apoptosis pathways to exert their antitumor functions.

### 2.6. Analysis of the Monocyte Differentiation during the Coculture with Cancer Cells

One of the most relevant questions to be addressed using this model is whether FaDu cells will modify the activation of monocytes and, if so, towards which type of macrophages. To answer this question, we targeted different markers specific to the polarization state, M1 (CD86), M2 (CD206), and pan-macrophage (CD68) markers, in our different conditions of cocultures. As shown in Figure 6A, CD68 marker was observed in all cocultured spheroids showing that the spheroid had been infiltrated throughout by monocytes, in the periphery and the necrotic core. M1–FaDu and M2–FaDu were used as positive controls of polarized macrophages. As expected, M1–FaDu spheroids induced by LPS and IFNγ expressed a higher level of the pro-inflammatory marker CD86 and a lower level of CD206, widely used to highlight the M2 subpopulation. On the other hand, M2–FaDu expressed more CD206 than CD86 (Figure 6A).

Interestingly, monocytes–FaDu spheroids revealed a high amount of CD206, especially in the periphery of the spheroid, and only a few CD86-positive cells. In addition, this condition also presented a higher level of CD206 than the M2-induced spheroids, suggesting that cancer cells were able to produce cytokines involved in the M2-like phenotype.

Next, a quantitative analysis of the CD206 cell surface marker in FaDu and monocytes–FaDu spheroids was performed by flow cytometry to confirm these results. Figure 6B,C demonstrate that the percentage of cells expressing the CD206-PE fluorescent emission reached 14% in monocytes–FaDu compared to 0.27% in FaDu alone, confirming the expression of CD206 by macrophages and a differentiation of monocytes into an M2-like phenotype initiated by cancer cells. Of note, 14% of CD206-positive cells in the monocytes–FaDu condition represent all macrophages that infiltrated the spheroids, as we reported a similar percentage of CD68-positive cells in the same coculture condition (as documented in Figure 6A).

### 2.7. Gene Expression Variations across the Spheroid Conditions

To further characterize macrophage phenotypes after 7 days of coculture with cancer cells, we quantified the gene expression of phenotypic and functional macrophage markers in FaDu and monocytes–FaDu dissociated spheroids compared to M1–FaDu and M2–FaDu used as controls (Figure 7).

Our results showed that the gene expression of the M2 phenotypic marker *CD206* was significantly increased in monocytes–FaDu compared to FaDu (*p* < 0.001), supporting that FaDu cells induced M2 polarization of monocytes as reported by immunofluorescence and flow cytometry. In addition, we also observed an increase in the mRNA expression of the anti-inflammatory cytokine *IL10* (*p* < 0.001). In fact, *IL10* secretion is reported as a marker for the M2 phenotype, further suggesting that infiltrating monocytes acquired such protumoral characteristics. As a positive control, these two markers were also expressed in M2–FaDu cells, further indicating that the monocyte profile differentiated towards an M2 phenotype. Furthermore, the diminished expression of the M1 markers *CD80* and *CD86*, which are typically associated with M1–FaDu cells, in the monocytes–FaDu group indicates that the monocyte phenotype does not align with the M1 phenotype.

Finally, mRNA expression of EMT markers of the cancer cells was also examined in the monoculture FaDu compared to the coculture monocytes–FaDu (Figure 8). We observed a clear decreased expression of the epithelial marker (*E-cadherin*) along with a significant increase in the mesenchymal marker (*Vimentin*) (*p* = 0.018) in the coculture compared to the FaDu monoculture. These last results suggested that monocytes differentiated in macrophages with cancer cells may promote the EMT process, as it is a reported function of M2 macrophages on cancer cells.

### 2.8. Cytokine Profile Variations in Culture Medium of 3D Spheroid

To investigate the impact of infiltrating monocytes in FaDu cancer cell spheroids on the cytokine secretion profile, we used the Human Cytokine Array Membrane to analyze the levels of 36 different cytokines and chemokines in the conditioned medium. The cytokines were measured on day 7 in FaDu cells with and without the addition of monocytes.

Our analysis revealed two significant observations. Firstly, the addition of monocytes led to a noteworthy increase in the level of CCL2, which was not observed in the supernatant of FaDu cells cultured alone (*p* = 0.002). This suggests that the presence of monocytes influences the secretion of CCL2 by the spheroids (coming from monocytes of cancer cells). Secondly, we observed a higher amount of IL8 in the supernatant of monocytes–FaDu cells compared to FaDu cells cultured alone (*p* = 0.014). This indicates that the interaction between monocytes and FaDu cells results in an elevated secretion of IL8 (Figure 9A,B).

To enhance the credibility of the results and validate the significance of the observed differences, we conducted RT-qPCR on CCL2 and IL8 (Figure 9C,D).

## 3. Discussion

Macrophages play a crucial role in cancer progression. As the tumor grows and develops, the cancer cells release signals that attract macrophages into the TME to promote their polarization towards the M2 phenotype. This polarization in turn leads to the production of cytokines and growth factors favoring tumor growth. Furthermore, through their immunosuppressive behavior, M2-polarized macrophages weaken the immune system, which allows cancer cells to evade detection and destruction by the immune system. Prior research has examined the relationship between TAMs and HNC, revealing that CD163+ TAMs are substantially elevated in such patients and are inversely correlated with recurrence-free survival (RFS), progression-free survival, and overall survival (OS) [9,31]. These findings suggest that targeting macrophages and especially the M2 phenotype may be a potential strategy for improving the prognoses and outcomes of head and neck cancer patients.

In this context, new research is needed to better understand the role of macrophages in cancer progression and to evaluate their use as prognostic markers. Nowadays, the evaluation of HNC patient outcomes is based on the grading of histopathology, the staging using TNM criteria, and the presence of HPV in the case of oropharyngeal cancer [18,32]. Nevertheless, there is evidence supporting the idea that the immune response within the TME plays an important role in determining patient outcomes. Previous studies have suggested that a high immunoscore may be associated with a better prognosis in head and neck cancer, while others have found no significant correlation [33,34,35]. Actually, the current screening methods are not fully adequate, and there is a pressing need to identify reliable biomarkers to quantify and understand the natural immune response in order to guide clinicians in cancer treatment decisions.

In our study, we counted the number of CD80-positive and CD163-positive cells in a training cohort of 54 HNC patients. Then, we established specific thresholds to differentiate between high and low macrophage infiltration for M1 and M2 subtypes, respectively. Using these cut-off values, we subsequently investigated the effect of macrophage counts, using ratios and summation on patient survival. Our study revealed significant variability in the recruitment of macrophages among HNC patients. This heterogeneity suggests that immune recruitment and activity may influence patient prognosis by differences in macrophage infiltration and phenotype across individuals. With this information, a novel Macroscore was calculated by combining the M2/M1 ratio and the quantity of M2 + M1 in HNC patients. This study demonstrated that patients with a high Macroscore had significantly poorer survival outcomes than those with a low score.

Next, to validate the significance of the prognostic value of the Macroscore, we utilized an independent validation cohort consisting of 19 patients. In this cohort, we quantified the expression levels of CD80 and CD163, and we applied our scoring system, established above, to this validation group. Our data validate that the Macroscore may define two groups of patients associated with low or long survival. Subsequently, we performed cross-tabulation analyses with the validation population and demonstrated that the Macroscore exhibits excellent predictive capabilities for differentiating between patients with high and low Macroscore values and predicting patient survival (dead or alive). Indeed, it exhibited a high specificity of 78% and a remarkable sensitivity of 90%. Furthermore, the PPV of 82% and NPV of 88% further underscored the reliability of the Macroscore in predicting positive and negative outcomes, respectively. These values indicate the probability of a positive or negative prediction being accurate based on the Macroscore assessment.

The M2/M1 macrophage ratio is a marker of the polarization of macrophages in the tumor microenvironment. This ratio could represent either a positive or a negative impact on tumor growth. The increased M2/M1 macrophage ratio is thought to promote tumor growth and invasion by creating an immunosuppressive microenvironment that allows cancer cells to evade the immune system. A higher M2/M1 ratio often indicates a poor prognosis in cancer patients, while a better prognosis is associated with a lower M2/M1 ratio [16]. Petrillo et al. showed that patients with high M1/M2 ratios had a greater response to chemotherapy or radiotherapy in locally advanced cervical cancers [36]. Another study, by Meiying et al., showed that ovarian cancer patients with increased overall or intra-islet M1/M2 ratios presented an improved 5-year survival [37]. Similarly, in gastric cancer, a higher M1/M2 ratio showed a better survival rate [38]. Hence, our results support these previous studies, showing that our Macroscore is a robust and unique tool that can effectively stratify HNC patients based on macrophage infiltration and phenotype, which is not achievable using conventional methods such as TNM staging or p16 status assessment. Notably, our patient cohort was highly heterogeneous, encompassing diverse tumor locations, stages, and patient characteristics such as smoking status. Despite this heterogeneity, the Macroscore exhibited strong prognostic performance. These results also highlight the valuable complementary information that the Macroscore can provide to enhance traditional diagnostic markers. By integrating the Macroscore with current factors/markers, clinicians can gain better insights and refine their assessment of HNC patient prognoses, ultimately contributing to more informed treatment decisions and improved patient care.

Understanding the intricate interactions between the TME and cancer cells is crucial for improving clinical management. Mimicking the in vivo conditions of the TME by using in vitro models is a challenge that may support a deep understanding of the interplay between immune and cancer cells, including the immune cell recruitment into tumors. Our clinical study has demonstrated that the majority of macrophages in the TME exhibit the M2 phenotype. Using a spheroid model, we cocultured head and neck cancer cells with undifferentiated monocytes and compared them to cocultures with monocytes priorly differentiated by the addition of any cytokines or growth factors. Our study using immunofluorescence and FACS demonstrated that the presence of cancer cells was necessary to recruit monocytes and generate spheroids, as monocytes or macrophages alone could not form spheroids. Furthermore, the cancer cells strongly exhibited a preference for inducing an M2-like macrophage phenotype over the M1 one. This polarization of monocytes towards the M2 phenotype was evident from the high levels of CD206 protein and mRNA expression, which are known markers associated with M2 macrophages. On the other hand, the expression of M1 markers such as CD80 and CD86 was found to be low in the cocultured monocytes–FaDu.

This result suggests that our 3D model successfully replicated an immunosuppressive microenvironment as evidenced by the larger growth of M2-infiltrated spheroids. Other studies have also employed spheroid models to investigate the effects of 3D cocultures on immune and cancer cells. For example, Janina Kuen found that coculturing pancreatic cells with fibroblasts led to the production of immunosuppressive cytokines, which promoted the M2 phenotype in monocytes. Additionally, when T cells were added to hybrid spheroids, M2 monocytes inhibited CD4+ and CD8+ T cell proliferation and activation [39]. Also, in a 3D model of pancreatic cancer cell/fibroblast coculture, Madsen et al. demonstrated that M2-like macrophages influenced therapeutic response [30]. Others, observing the infiltration of monocytes in the spheroids, reported that their differentiation into mature macrophages with diverse phenotypes depended on the cancer cell line [30]. Raghavan et al. used PBMCs or cell line-derived monocytes to show that ovarian cancer spheroids cocultured with monocytes were more invasive and less sensitive to chemotherapy in vitro [40], while others demonstrated the presence of M2 macrophages in a heterotypic spheroid and explant model of early-stage lung cancer [41]. These studies collectively highlight the importance of incorporating the TME into 3D models to investigate the complex interactions between immune cells and cancer cells.

In addition, by using our model, we were able to distinguish between two crucial macrophage phenotypes and their functions. In our study, we formed spheroids infiltrated with monocytes converted to CD86+ M1 macrophages by LPS and IFNγ or monocytes differentiated to CD206+ M2 ones by the addition of IL4 and IL13 cytokines. Consistent with spheroid size measurements and Ki-67 immunostaining, we found that the M2 macrophages increased the spheroid size with a high percentage of proliferation. Conversely, the addition of M1 macrophages resulted in a decrease in both spheroid size and proliferation rate. These findings are also reported by Yuan A et al. in lung cancer, showing that M2 macrophages increased spheroid volume while M1 macrophages decreased it compared to spheroids formed from cancer cells alone [42]. In addition, our findings revealed a contrast in the apoptotic profiles of M1 and M2 macrophages, with M1 demonstrating a higher percentage of cell apoptosis within the spheroid compared to M2.

After validation of the M2 phenotype induced by cancer cells in our 3D model, we wanted to investigate the mechanisms underlying this phenomenon, and we analyzed the profile of cytokines secreted in the culture medium conditioned by cancer cells with or without monocytes. An increase in the expression of CCL2 and IL8 was observed when cancer cells were cocultured with monocytes. Of note, macrophages, monocytes, and dendritic cells primarily secrete CCL2, also known as MCP-1. This protein allows the recruitment of macrophages during acute inflammation by binding to its receptor CCR2, which can regulate the immunoreactivity of the TME. The interaction between CCL2 and CCR2 is essential for macrophage-related functions in cancer progression, such as mediating cancer-related inflammation, regulating the balance between M1 and M2 macrophages, facilitating the recruitment of TAMs, and providing anti-apoptotic signals [43]. A recent study showed that the activation of the CCL2–CCR2 axis promotes the recruitment of TAMs into the TME in several tumor types, like retinoblastoma [44].

On the other hand, IL-8 is a pro-inflammatory interleukin which is involved in tumor growth, invasion, and metastasis. A systematic review and meta-analysis found that high IL-8 expression was significantly associated with poor overall survival and short disease-free survival in patients with head and neck cancer [45]. In addition, Kazuki Kai et al. showed that oral squamous cell carcinoma (OSCC) promotes the differentiation of monocytes to CD206+ TAMs via IL-8 production. Moreover, the number of IL-8+ cells is significantly associated with the progression-free survival rate [46]. Another study found that macrophage-derived IL-8 promotes colon cancer cell invasion and migration. The study showed that inhibiting IL-8 signaling in macrophages reduced colon cancer cell invasion and migration in vitro and inhibited colon cancer metastasis in vivo. Moreover, Qiaoshi Xu et al. showed that IL-8 can initiate the EMT program to promote the malignant progression of HNC [47], as we observed in our 3D spheroid model. Indeed, using recombinant human IL-8 as a treatment of head and neck cancer cells, their Western blot analyses revealed a decrease in the expression of E-cadherin and an increase in the expression of MMP2, MMP9, and vimentin compared to the untreated conditions. Similarly, our IL-8 overexpressing coculture conditions (Human Cytokine Array) also seemed to promote an EMT transition with a significant increase in vimentin expression along with a trend of decrease in E-cadherin level, as demonstrated by RT-qPCR. These results suggested that M2 macrophages induced the EMT mechanism in cancer cells. EMT is a biological process that is involved in the progression of cancer. During EMT, cancer cells lose their epithelial characteristics and acquire mesenchymal features, which allows them to invade surrounding tissues, migrate to distant sites, and resist cancer treatment. Lu Gao et al. demonstrated that TAMs can induce the EMT of HNC cells by secreting EGF and TGF-β, resulting in an increased invasive ability of the cancer cells [48]. In addition, Yong Hu et al. showed that the CD163 M2 marker has a significant relationship with E-cadherin and vimentin expression, indicating that such TAMs are involved in the EMT. Another study involved the activation of the NF-κB pathway, which can induce the expression of genes promoting EMT [10]. Supporting these data, Qi Zhanga et al. showed that the use of the NF-κB inhibitor prevents the presence of TAMs and suppresses EMT [49]. These findings allow us to gain more insights into the critical bidirectional relationship between the tumor microenvironment and cancer phenotype.

Additionally, a significant upregulation of the IL10 anti-inflammatory cytokine was observed in the monocytes–FaDu spheroid by our RT-qPCR experiments. IL-10 is well-known for its role in M2 macrophage differentiation and for its immunosuppressive effects, which may contribute to tumor growth and progression [50]. In pancreatic cancer, studies showed that the presence of IL10 has been associated with poor prognosis [51]. In addition, Giri et al. showed that CCL2 can upregulate IL10 expression in CCR2+ macrophages [43], showing that several regulatory loops exist between cytokines.

## 4. Materials and Methods

### 4.1. Patients and Clinical Characteristics

Formalin-fixed paraffin-embedded HNC specimens were obtained from 54 patients who underwent curative surgery at Saint-Pierre Hospital (Brussels, Belgium) and the Jules Bordet Institute (Brussels, Belgium) during the years 2010 to 2021. This cohort was used as the training population (Table 1). The validation cohort consisted of 19 independent patients and was obtained at Saint-Pierre Hospital during the years 2020 to 2021 (Table 6). This retrospective study was approved by the Institutional Research Ethics Board of the Jules Bordet Institute (number CE2319). Clinicopathological characteristics of the patients are summarized in Table 1 and Table 6, which present information concerning the patients’ age, gender, tumor localization, tumor stage, risk factors including p16 positivity, clinical follow-up, and overall survival (OS).

### 4.2. Immunohistochemistry and Macrophage Number Quantification

All 73 FFPE specimens were labeled by immunohistochemistry for CD163 and CD80 immune markers to identify M2 and M1 macrophage populations, respectively. The protocol was fully described in a previous publication [52]. To determine the transcriptional status of HPV, all specimens were further immunohistochemically evaluated for p16 expression using a mouse monoclonal antibody (CINtec p16, Ventana, Tucson, AZ, USA) and an automated immunostainer at the Jules Bordet Institute (Bond-Max, Leica Microsystems, Wetzlar, Germany), as described previously [53].

Regarding the macrophage number quantification, for the training cohort, the immune cell count in IT and ST compartments was performed by two investigators (N.M. and G.D.) using an Axio-Cam MRC5 optical microscope at 400× magnification, as explained in a previous publication [33]. For the validation cohort, the quantification was performed using QuPath software 0.4.3 version [54], which involved counting the number of positive cells per square millimeter (mm^2^) in both intratumoral and stromal regions of the tissue. For both populations, RStudio software (V 4.2.3) was used to determine the optimal cut-off values for each marker, giving the best separation between the two groups based on the hazard ratio and *p*-value for OS. Cases with mean density greater than the cut-off were classified as “high”, while those with lower mean density were classified as “low”. The prognostic value of each immune marker was then assessed for OS.

### 4.3. Cell Culture

The FaDu human HNC cell line (ATCC^®^ HTB-43, Manassas, VA, USA) was cultured in RPMI (Roswell Park Memorial Institute Medium 1640, Lonza, Basel, Switzerland) containing 10% heat-inactivated fetal bovine serum (FBS Premium South America, PAN BIOTECH, Aidenbach, Germany); 5% L–glutamine (200 mM, Gibco, Thermo Fisher Scientific, Waltham, MA, USA); and 1% penicillin/streptomycin (10,000 U/mL/10,000 μg/mL, Gibco, Thermo Fisher Scientific) and maintained at 37 °C under 5% CO_2_. The culture medium was replaced every 2 days, and subcultures were processed by using Trypsin-EDTA (PAN^TM^Biotec, Darmstadt, Germany) when cells reached 80% of confluence. Cell lines were mycoplasma-free and tested regularly.

### 4.4. PBMC Purification and Isolation

The buffy coat received from a healthy donor (Red Cross, Suarlée, Belgium) was used for monocytes isolation. Peripheral blood mononuclear cells (PBMCs) were separated based on the density gradient of Ficoll™ 1.077 g/mL (Lymphosep BioWest, Nuaillé, France) and centrifuged for 20 min at 900× *g* without deceleration. Three phases were formed after centrifugation: erythrocytes and granulocytes settled at the bottom of the tube and the platelets in the supernatant, while monocytes and lymphocytes formed a ring between the two phases. This ring was collected, washed with HBSS (Hank’s Balanced Salts Solution, Gibco), and centrifuged for 10 min at 450× *g* with a deceleration of 9. Then, PBMCs were resuspended with FBS/DMSO 10% and counted in order to be stored at a density of 25 million per cryotube.

Then, CD14+ magnetic microbeads (Miltenyi Biotec, Leiden, The Netherlands) and separation columns (Miltenyi Biotec, Leiden, The Netherlands) were used to isolate the CD14+ monocytes with MiniMACS Separator (Miltenyi Biotec, Leiden, The Netherlands) equipment. The primary monocytes obtained were used for the establishment of the coculture.

### 4.5. Tumor Spheroid Formation and Characterization

Spheroids were generated by the liquid overlay technique (LOT) using U-bottom 96-well plates (Brand 96-W Brandplates U-bottom) with an ultra-low cell adhesion surface promoting the aggregation of cells. They were performed in either monoculture or coculture from a cell suspension containing the required number of cells, which was dispensed into the 96-well plate (100 μL/well) according to four culture conditions. Four cultures conditions were realized: (1) Monoculture condition composed of 500 cells of FaDu, which served as the control. (2) Coculture condition composed of a mixture of CD14+ monocytes isolated from PBMCs and FaDu cancer cells in a 4:1 ratio, which was generated without addition of any stimuli. (3) Coculture of M1 macrophages and FaDu cancer cells condition generated by the addition of 10 pg/mL LPS (lipopolysaccharides from Escherichia Coli O111:B4, Merck L2630-10 mg, Hoeilaart, Belgium) and 20 ng/mL INF-γ (Recombinant Human IFN-gamma Protein, R&D Systems 285-IF-100, Minneapolis, MN, USA), which polarize monocytes in M1 macrophages. (4) Coculture of M2 macrophages and FaDu cancer cells condition generated by the addition of 20 ng/mL IL-4 (human interleukin-4 protein 20 μg, ELL172, Interchim, Montluçon, France) and IL-13 (human interleukin-13 protein 20 μg, ELD140, Interchim, Montluçon, France), which polarize in M2 phenotype. The spheroid cultures were maintained for 7 days with a renewal of the media on day 3. Brightfield images were captured on days 2, 5, and 7 through an inverted microscope coupled with a camera (EUROMEX HD) to assess their growth evolution. Finally, the volume of spheroids was analyzed with Image J. First, the pixel area of each spheroid was measured with Image J, and then the pixel radius was calculated using the formula for the area of a sphere ($A=\pi r^{2})$. Next, a known distance in mm was measured in pixels, with 0.25 mm equaling 396.0455 pixels. A proportional calculation was then applied to convert the radius from mm to pixels. Based on the radius in mm, the volume (mm^3^) of the sphere was calculated ($V=\frac{4}{3}\pi r^{3}$).

### 4.6. Immunofluorescence

On the 7th day of coculture, spheroids were washed in PBS and fixed with paraformaldehyde 4% (Sigma-Aldrich, St. Louis, MI, USA) overnight at 4 °C and then rinsed in PBS. A permeabilization step is required overnight with 0.5% triton X-100 at 4 °C followed by washing steps with PBS. Next, spheroids were incubated with a blocking solution for 2 h followed by antibody incubation for 24 h (Table 7). The next day, spheroids were incubated with the secondary antibody (1/500 dilution) for 1 h followed by rinsing with PBS and distilled water. Then, Hoechst dye (BisBenzimide Sigma Aldrich) was used for 15 min in order to stain nuclei in spheroids. Finally, they were transferred onto a coverslip and mounted on the slides with Vectashield (Vector laboratories, Newark, CA, USA). Once they were dried, slides were observed under a confocal microscope (Nikon Ti2 A1RHD25, Tokyo, Japan). Quantification of fluorescence intensity was examined with QuPath software Version 0.4.3.

### 4.7. RNA Extraction, cDNA Synthesis, and qPCR

First, spheroids were harvested, pooled (*n* = 192), and centrifuged for 5 min at 1200 rpm. Then, spheroids were digested with Accutase (Gibco StemPro Accutase) for 15 min with pipetting up and down every 5 min until total dissociation. RNA extraction was performed on cell pellets with the InnuPrep RNA mini kit 2.0 (Annalytik Jena, Jena, Germany) according to the manufacturer’s protocol. Quantification and purity of the isolated RNAs were determined using the nanodrop (Bio-Drop μlite, Fisher scientific). Then, 1 or 2 µg of RNA was used for retrotranscription` in cDNA using the Maxima First Strand cDNA Synthesis Kit for RT-qPCR with dsDNase (Thermo Scientific, K1671, Waltham, MA, USA). Purity of cDNAs was also verified with the nanodrop, and then they were diluted 10 times before being analyzed by quantitative real-time polymerase chain reaction (RT-qPCR). The reaction mixture composed of SYBR Green Mix (Takyon Rox SYBR Core Kit Blue dTTP, Eurogentec, Selland, Belgium), 10 µM of the forward and reverse primers (produced by IDT, Integrated DNA Technologies, Leuven, Belgium) (Table 8), and RNAse-free water was dispensed in a 96-well plate (Microplate for PCR, 96 wells, with sealing film, 732–1591 VWR), and then cDNAs were added to each well in triplicate. Then, the plate was introduced into a thermocycler (LightCycler 96 FW13083, Roche, Bâle, Switzerland). The reaction process followed several steps: The first consisted of the denaturation of DNA for 10 min at 95 °C; the second consisted of amplification by 40 cycles for 15 s at 95 °C and 1 min at 60 °C; and the final consisted of a melting curve of 15 s at 95 °C, 1 min at 62 °C, followed by continuous acquisition at 95 °C. Data were analyzed with LightCycler^®^ 96 SW 1.1 software, and the relative expression of the genes was calculated (2^−ΔCt^) and normalized according to the level of the housekeeping gene 18S.

### 4.8. Flow Cytometry

FaDu and monocytes–FaDu spheroids were dissociated as described above, and 5 × 10^4^ cells were centrifuged for 5 min at 1200 rpm and washed with DPBS/FBS 5%. Staining for the surface marker CD206–APC (anti-human, monoclonal recombinant IgG1, DCN228, Miltenyi Biotec, Leiden, The Netherlands) was performed for 10 min at 4 °C in the dark at a 1/50 dilution in DPBS/FBS 5%. After washing, the cells were resuspended in DPBS/FBS 5% and analyzed by flow cytometry. The samples were assessed in Fluorescence-Activated Cell Sorting (Beckman Coulter NaviosTM) and the data were analyzed using Kaluza software version 2.1

### 4.9. In Vitro Apoptosis Assay

Apoptosis of FaDu, monocytes–FaDu, M1–FaDu, and M2–FaDu dissociated spheroids was determined using the Annexin V & Dead Cell Kit (Muse^®^ Annexin V & Dead Cell Kit, Luminex, Austin, USA). A cell suspension of 5 × 10^5^ cells/mL was incubated with 100 μL of Muse reagent Annexin V for 20 min at room temperature. Then, the solution was vortexed and analyzed by a Muse flow cytometer (Guava^®^ Muse^®^ Cell Analyzer, Luminex, Darmstadt, Germany) and the percentages of early and late apoptosis cells as well as living cells were quantified.

### 4.10. Assessment of Cytokine Profiles

Supernatants of FaDu and monocytes–FaDu spheroids were collected at the 7th day of coculture. Samples were centrifuged for 5 min at 1200 rpm to remove debris and particles and stored at −20 °C until use. Quantification of cytokines in supernatants was evaluated using a Proteome Profiler Human Cytokine Array Kit (R&D Systems ARY005B, Minneapolis, MN, USA). The kit contains 36 cytokines spotted on a nitrocellulose membrane. Supernatants were mixed with a cocktail of biotinylated detection antibodies according to the manufacturer’s protocol. Streptavidin–horseradish peroxidase and chemiluminescent detection reagents were added, and a signal was revealed that is proportional to the amount of cytokine bound. Chemiluminescence was detected by autoradiography with the FUSION FX (Vilber, Collégien, France). Protein quantification was performed by analyzing pixel density in each spot using ImageJ software version 1.54d

### 4.11. Statistical Analysis

Data presented are representative of three independent experiments (*n* = 3). Statistical analyses were performed using IBM SPSS software (version 21) (IBM, Ehningen, Germany). For clinical study, OS analyses were performed using Kaplan–Meier curves. Univariate and multivariate Cox regression models were applied to calculate the hazard ratio (HR) and significance. The specificity, sensitivity, positive predictive value (PPV), and negative predictive value (NPV) of the Macroscore were evaluated from crosstabs based on cut-off points, and significance was calculated using the Chi-square test. For in vitro experiments, differences between experimental independent groups were compared using a *t*-test or ANOVA test followed by a Tukey’s post hoc test. A *p*-value < 0.05 was considered as significant (* = *p* ≤ 0.05; ** = *p* ≤ 0.01; *** = *p* ≤ 0.001).

## 5. Conclusions

TAMs play a crucial role in promoting tumor growth by secreting factors that stimulate tumor cell proliferation and immunosuppression. Altogether, our research reports the identification of CD80 and CD163 markers as useful for the creation of a novel scoring system (Macroscore) based on the M2/M1 ratio and the M2 + M1 summation in patients suffering from HNC. This Macroscore is effective in categorizing HNC patients based on macrophage infiltration and phenotype and can provide valuable information for patient prognosis and treatment decisions. Additionally, we developed a 3D coculture model based on spheroid formation that mimics in vivo conditions and creates an immunosuppressive microenvironment, allowing us to explore the interplay between immune and cancer cells. Our findings indicate that cancer cells can recruit monocytes through the secretion of CCL2 cytokine and induce their differentiation into the M2 phenotype. Reciprocally, M2 macrophages can also produce cytokines such as IL10 and IL8, which promote tumor progression and the epithelial–mesenchymal transition.

## Figures and Tables

**Figure 1 ijms-24-12813-f001:**
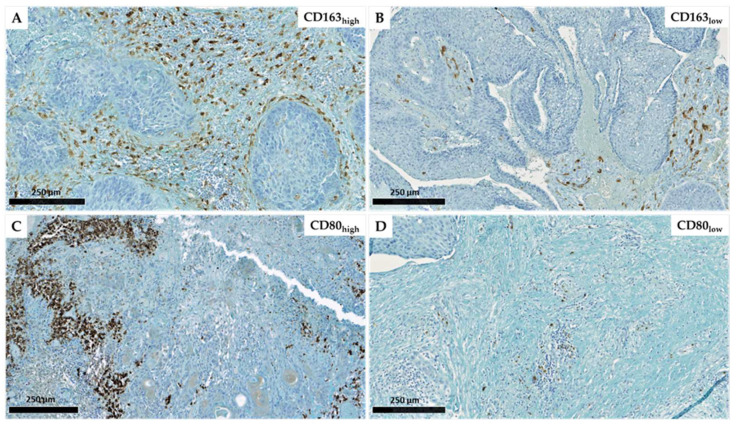
Immunohistochemical representation of CD163 expression with a high (**A**) and low (**B**) density as well as CD80 high (**C**) and low (**D**) expression in head and neck carcinomas (scales = 250 µm).

**Figure 2 ijms-24-12813-f002:**
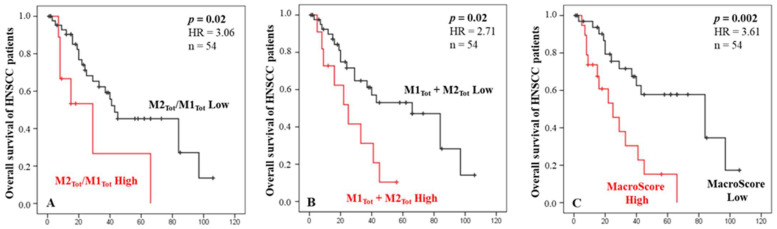
Kaplan–Meier curves of the OS of HNC patients according to the macrophages ratio score (**A**), the total amount of macrophages M1 and M2 infiltrating the tumor and stroma (**B**), and the Macroscore combining the ratio and quantity of macrophages (**C**). X axes are time in months.

**Figure 3 ijms-24-12813-f003:**
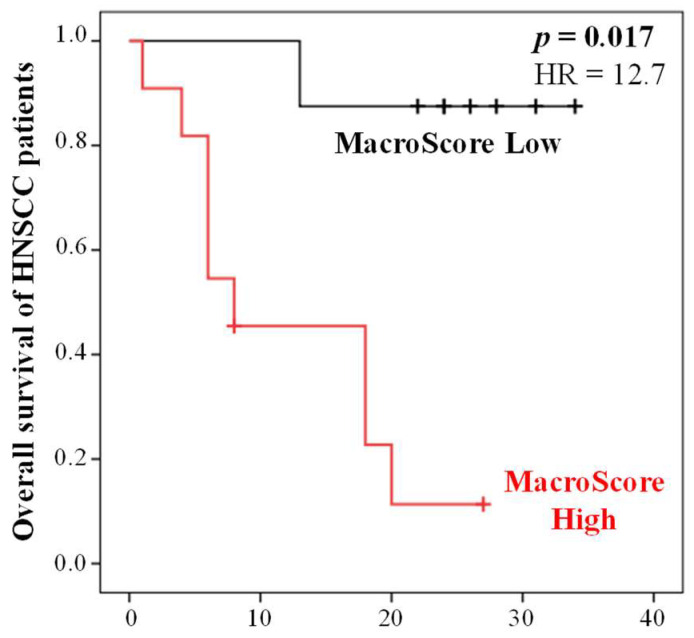
Kaplan–Meier curves of the OS of HNC patients (validation cohort, *n* = 19) according to the Macroscore defined using the training population. X axis is time in months.

**Figure 4 ijms-24-12813-f004:**
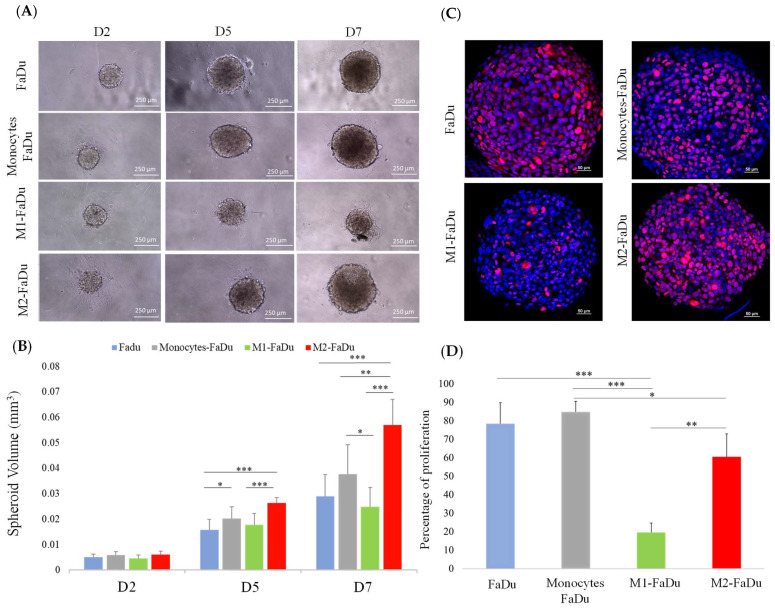
Coculture of FaDu cells and macrophages in spheroids. (**A**) Photographs of spheroids containing cancer cells with or without macrophages. (**B**) Quantification of spheroid volumes for the different coculture conditions (scale = 250 μm). (**C**) Comparison of Ki-67 proliferation marker expression by immunofluorescence in the different coculture conditions on day 7. (**D**) Quantitative data of Ki-67 expression performed with QuPath. Mean + SD, ANOVA one-way, and Tukey’s post hoc test (* *p* ≤ 0.05; ** *p* ≤ 0.01; *** *p* ≤ 0.001), *n* = 8.

**Figure 5 ijms-24-12813-f005:**
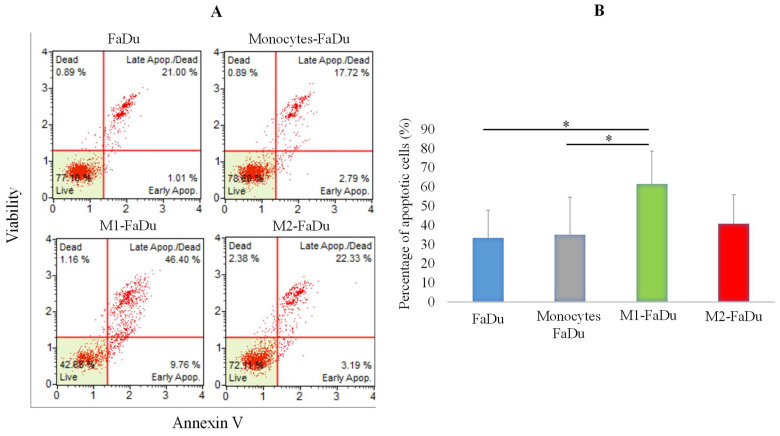
Evaluation of apoptosis in coculture spheroids. (**A**) Graphs showing raw data from flow cytometer assessing annexin V staining vs. cell viability. (**B**) Quantification of the number of apoptotic cells within spheroids of the different coculture conditions using the annexin V/dead cell marker assay. Mean ± SD, ANOVA one-way and Tukey’s post hoc test (* *p* ≤ 0.05), *n* = 3.

**Figure 6 ijms-24-12813-f006:**
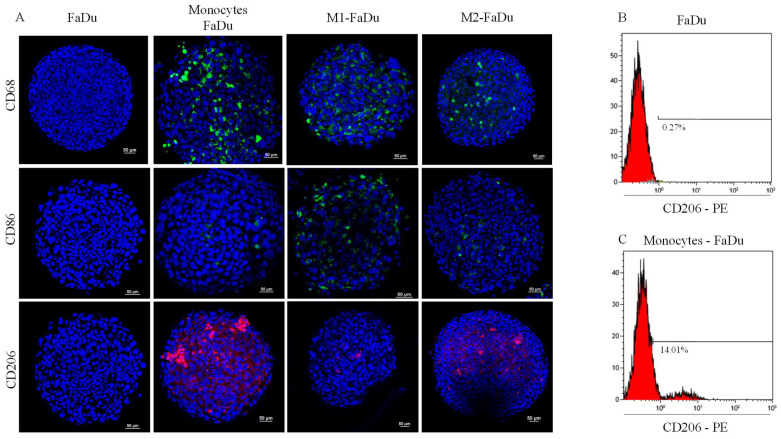
Effect of cancer cells on monocyte polarization. (**A**) Evaluation of the expression of macrophage phenotypic markers: CD68, CD86, and CD206 on day 7 in the different coculture conditions (scale = 50 μm); M1–FaDu and M2–FaDu were used as positive control. (**B**,**C**) FACS analyses to quantify the percentage of M1 and M2 markers in FaDu–monocyte dissociated spheroids. Dot plots of CD206 expressed on FaDu spheroids, used as a negative control (no macrophage), and in monocytes–FaDu spheroids.

**Figure 7 ijms-24-12813-f007:**
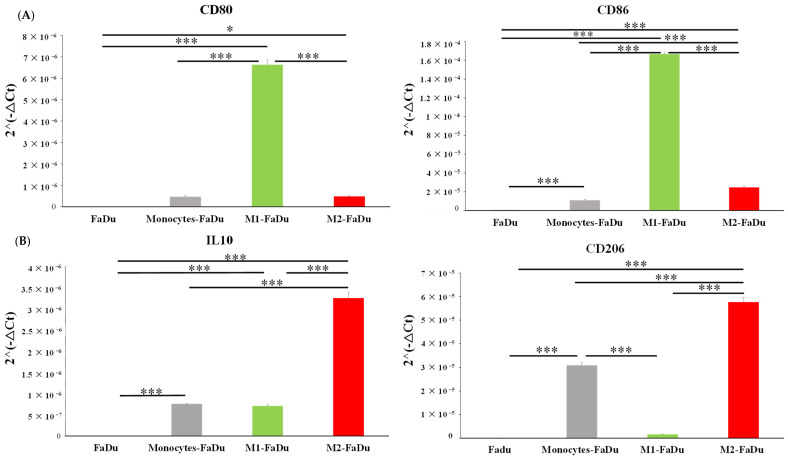
mRNA relative expression (2^−ΔCt^) to characterize macrophage phenotype using (**A**) M1 markers (CD80, CD86) and (**B**) M2 markers (IL10, CD206). The analyses by RT-qPCR and the normalization with 18S expression were conducted on dissociated FaDu and monocytes–FaDu, M1–FaDu, and M2–FaDu spheroids. Mean + SD, *t*-test, * = *p* ≤ 0.05, *** *p* ≤ 0.001.

**Figure 8 ijms-24-12813-f008:**
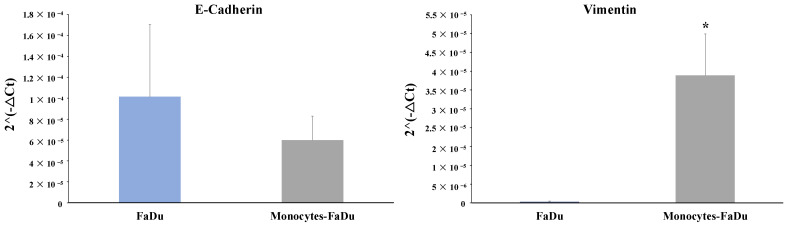
mRNA relative expression (2^−ΔCt^) to characterize EMT markers (E-cadherin and Vimentin). The analyses by RT-qPCR and the normalization with 18S expression were performed on dissociated FaDu and monocytes–FaDu spheroids. Mean + SD, *t*-test, * = *p* ≤ 0.05.

**Figure 9 ijms-24-12813-f009:**
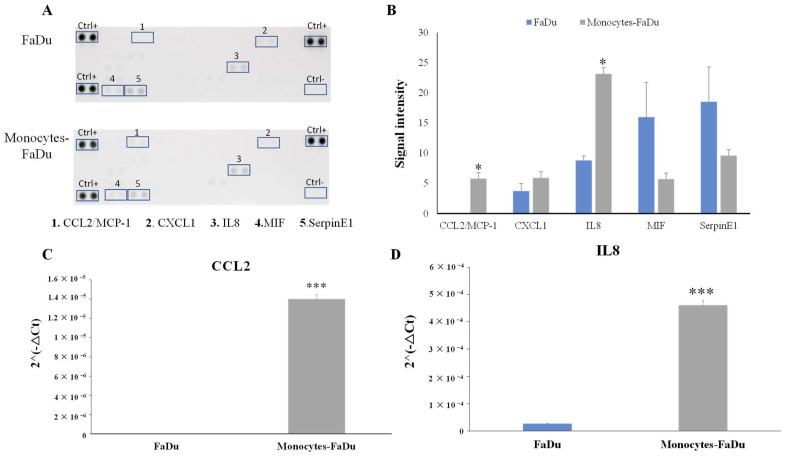
Human Cytokine Array Membrane (**A**,**B**) of supernatant of FaDu cancer cell spheroids and monocytes–FaDu spheroids on day 7. (**A**) Membranes. (**B**) Quantification showing the mean signal intensity (*n* = 3) normalized to positive control (Ctrl+). mRNA relative expression (2^−ΔCt^) of CCL2 and IL8 cytokines (**C**,**D**). Analyses by RT-qPCR and normalization with 18S expression on dissociated FaDu and monocytes–FaDu spheroids. Mean + SD, *t*-test, * = *p* ≤ 0.05, *** *p* ≤ 0.001.

**Table 1 ijms-24-12813-t001:** Clinical patient characteristics (training population).

Variables	Number of Cases
	*n* = 54
**Age**	
Median (range, years)	62 (42–89)
**Gender**	
Male	41
Female	13
**Anatomic site**	
Oral cavity	20
Oropharynx	16
Larynx	15
Hypopharynx	2
Nasopharynx	1
**Tumor stage**	
I–II	31
III–IV	16
Unknown	7
**Risk factors**	
*Tobacco*	
Smoker	47
Non-smoker	7
*Alcohol*	
Drinker	33
Non-drinker	21
*p16 status*	
Positive	26
Negative	28
**Overall survival (OS)**	
Median (range, months)	24 (1–106)
Alive	26
Dead	28

**Table 2 ijms-24-12813-t002:** Multivariate analysis evaluating the correlation between M2_Tot_/M1_Tot_ score, M1_Tot_ + M2_Tot_ score, and patient survival. *p*-values < 0.05 are highlighted in bold.

Multivariate Analysis	Overall Survival	
	*p*-Value	HR
M2_Tot_/M1_Tot_ score	**0.005**	4.19
M1_Tot_ + M2_Tot_ score	**0.005**	3.47

**Table 3 ijms-24-12813-t003:** Multivariate analysis evaluating the correlation between M2_Tot_/M1_Tot_ score, M1_Tot_ + M2_Tot_ score, p16 status, and patient survival. *p*-values < 0.05 are highlighted in bold.

Multivariate Analysis	Overall Survival	
	*p*-Value	HR
M2_Tot_/M1_Tot_ score	**0.004**	4.31
M1_Tot_ + M2_Tot_ score	**0.014**	3.14
p16	0.525	0.76

**Table 4 ijms-24-12813-t004:** Multivariate analysis evaluating the correlation between M2_Tot_/M1_Tot_ score, M1_Tot_ + M2_Tot_ score, tumor staging, and patient survival. *p*-values < 0.05 are highlighted in bold.

Multivariate Analysis	Overall Survival	
	*p*-Value	HR
M2_Tot_/M1_Tot_ score	**0.007**	4.73
M1_Tot_ + M2_Tot_ score	**0.009**	3.64
Stage	0.728	0.84

**Table 5 ijms-24-12813-t005:** Multivariate analysis evaluating the correlation between the Macroscore, staging, p16, and patient survival. *p*-values < 0.05 are highlighted in bold.

Multivariate Analysis	Overall Survival	
	*p*-Value	HR
Macroscore	**0.006**	3.81
p16	0.886	1.07
Stage	0.874	0.92

**Table 6 ijms-24-12813-t006:** Clinical characteristics of patients (validation cohort).

Variables	Number of Cases
	*n* = 19
**Age**	
Median (range, years)	59 (31–72)
**Gender**	
Male	15
Female	4
Anatomic site	
Oral cavity	11
Oropharynx	2
Larynx	5
Hypopharynx	1
**Tumor stage**	
I–II	4
III–IV	15
**Risk factors**	
*Tobacco*	
Smoker	11
Non-smoker	3
Former smoker	4
*Alcohol*	
Drinker	5
Non-drinker	12
Former drinker	1
*p16 status*	
Positive	1
Negative	18
**Overall survival (OS)**	
Median (range, months)	18 (1–34)
Alive	10
Dead	9

**Table 7 ijms-24-12813-t007:** Table representing the different conditions used in immunofluorescence.

Targets	Antibodies	Antibody Dilutions	Blocking Solutions	Secondary Antibodies
CD68	Anti-CD68, Anti-human, Rabbit monoclonal, Cell signaling (Danvers, MA, USA)	1/800	PBS/NGS 5%/Triton 0.3%	Goat anti-Rabbit IgG (H + L) Highly Cross-Absorbed Secondary Antibody, Alexa Fluor Plus 488
CD86	Anti-CD86, Anti-human, Rabbit monoclonal, Cell Signaling	1/100	PBS/NGS 5%/Triton 0.3%	Goat anti-Rabbit IgG (H + L) Highly Cross-Absorbed Secondary Antibody, Alexa Fluor Plus 488
CD206	Anti-CD206, Anti-human, Mouse monoclonal, Cell signaling	1/100	PBS/BSA 2%	Chicken anti-Mouse IgG (H + L) Highly Cross-Absorbed Secondary Antibody, Alexa Fluor 594
Ki-67	Anti-Ki-67, Anti-human, Mouse monoclonal, Cell signaling	1/200	PBS/NGS 5%/Triton 0.3%	Chicken anti-Mouse IgG (H + L) Highly Cross-Absorbed Secondary Antibody, Alexa Fluor 594
EpCAM	Anti-EpCAM, Anti-human, Goat monoclonal, Cell signaling	1/50	PBS/BSA 2%	Donkey anti-Goat IgG (H + L) Highly Cross-Absorbed Secondary Antibody, Alexa Fluor 647
Vimentin	Anti-Vimentin, Anti-human, Mouse monoclonal, Agilent (Santa Clara, CA, USA)	1/50	PBS/BSA 2%	Goat anti-Mouse IgG (H + L) Highly Cross-Absorbed Secondary Antibody, Alexa Fluor Plus 555

**Table 8 ijms-24-12813-t008:** List of qPCR primers.

Genes	Forward Sequences	Reverse Sequences
18S	CATTTAGGTGACACTATAGAAGACGATCAGATACCGTCGTAGTTCC	GGATCCTAATACGACTCACTATAGGCCTTTAAGTTTCAGCTTTGCAACC
IL10	TCAAGGCGCATGTGAACTCC	GATGTCAAACTCACTCATGGCT
CD206	CTACAAGGGATCGGGTTTATGGA	TTGGCATTGCCTAGTAGCGTA
E-cadherin	ATTTTTCCCTCGACACCCGAT	TCCCAGGCGTAGACCAAGA
Vimentin	AGTCCACTGAGTACCGGAGAC	CATTTCACGCATCTGGCGTTC
CD80	GGGCACATACGAGTGTGTTGT	TCAGCTTTGACTGATAACGTC AC
CD86	CTGCTCATCTATACACGGTTACC	GG AAACGTCGTACAGTTCTGTG

## Data Availability

Data are contained within the article or Appendix A.

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
