# Peer review of "Macrophage Profiling in Head and Neck Cancer to Improve Patient Prognosis and Assessment of Cancer Cell–Macrophage Interactions Using Three-Dimensional Coculture Models"

_ijms, 2023, doi:10.3390/ijms241612813_

Round 1
Reviewer 1 Report
The present manuscript by Nour et al investigated the prognostic potential of Macroscore based on the density of M1 and M2 macrophages in patients with head and neck cancer. The results demonstrated that Macroscore was significantly associated with poor patient prognosis. The use of 3D spheroids enabled the analysis of cancer cell-macrophage interactions, revealing that cancer cells can induce monocyte differentiation into M2 macrophages and create an immunosuppressive microenvironment.
The study is well-designed. They used a coculture spheroid model to better understand cancer cell-macrophage interactions.
However, some details need to be changed and more information needs to be provided.
I suggest the following specific comments and suggestions to improve the paper:
1. Can the authors provide details on how they determined the p16 status in their study?
2. The authors may consider including figures to show the formation of the spheroids in the results section 2.2.
3. In the result section 2.3, it would be beneficial for readers to know what specific types of monocytes were used in this study.
4. On line 487, it appears that "INF-GAMMA" should be corrected to "IFN-gamma."
5. How was the size of the spheroids measured? It would be nice if you could show any formulas in materials and methods.
6. Have they considered the effects of IFN-gamma, LPS, IL4, and IL13 on Fadu cells?
7. Please double-check the zoom on your figures to ensure the size is accurately represented. Additionally, the nuclear size appears to be different between cells.
8. line 416-417, it may be beneficial for the authors to discuss these findings in more detail to improve the understanding of the importance of Figure 7.
Additionally, some tables, such as the information on primers and antibodies, may be useful to include in the supplementary materials.
Reviewer 2 Report
The paper is very interesting, it focuses on a hot topic regarding TME and macrophages. The author calculated in HNC the M2/M1 ratio (as already published in other solid tumors) but they add the sum of M2 and M1 and associated these two indicators in a macroscore.A high macroscore is associated with high score.
I would add if possible a relation with different biomarkers such as LNI,TMB ,IFN signature
Reviewer 3 Report
1. In the introduction part, in line 60-66, the authors have that proinflammatory cytokines leads to anticancer immunity and anti-inflammatory cytokines have enhances cancer progression. However, the general concept is just the opposite (doi: 10.3389/fonc.2023.1133013). The authors must provide a detailed clarification in the introduction.
2. In line 107, what is the rationale or reference for the cut-offs?
3. Has the calculation of “macroscore” have any scientific reference?
4. Kindly provide the reference for “the best illustration of infiltration of immune cells in the tumor”.
5. In line 164, the authors have suggested that as the monocytes come in contact with FaDu cells they become polarized (i.e. become M1 or M2, of what I understand). So how are the three set of spheroids distinguished for 7 days? The pro-inflammatory and anti-inflammatory markers that are added in spheroids C and D could be the reason for the outcome alone. Is there any reference of the details of the concentration of these inflammatory markers that has been added?
6. In apoptosis assay, the experiments was performed in coculture, since M1 group had least spheroid growth, this could also mean that the ratio of monocytes and FaDu cells were very high. The high rate of apoptosis could also be due to apoptosis of monocytes, is there any way to differentiate the apoptosis of FaDu cells from that of monocytes. This is important because the other groups have low ratio of monocytes and FaDu cells, in control there are no monocytes at all, so it could be that the apoptosis is of mainly monocytes and less of FaDu cells.
7. Line 496, there is a typo error after “7th….”
8. The comparative gene expression studies were performed for FaDu cells and FaDu cells + monocytes. We would like to know why gene expression studies of the M1+FaDu, and M2 + FaDu cells were not performed?
English language is satisfactory.
Round 2
Reviewer 1 Report
1. Based on the scale bar provided in Supplementary Figure 1, the depicted images do not correspond to the results mentioned in line 160 of the manuscript. 2. In Figure 5A, it appears that there might be a discrepancy in the results compared to Figure 3A, assuming the author has verified the accuracy of the scale bar measurement. 3. In Figure 7, would greatly enhance the interpretation and statistical significance of the results of the quantification of the chemokine array provided for three independent replicates. 4. The response to Question 6 has left me more confused regarding the spheroid size and the number of cells that were seeded. It would be beneficial if the units used in the paper and within the answer could be checked.Author Response
Please see the attachment.

Reviewer 3 Report
1. The authors have not provided any scientific reference for calculation of macroscore, according to their response, it seems that they have come up with a new terminology “macroscore”, which may require experimental validation. It is extremely important that the authors highlight this significant experimental assumption in the methodology or else the research paper would have a significant flaw in though the outcome of the paper may be corroborated with other research publications with similar outcome.
2. We have not received any satisfactory response regarding query no. 8. The response of the authors does not highlight the significance of gene expression studies in the context of the objective of the research manuscript.
The English language is fair enough to provide scientific clarity.
Round 3
Reviewer 1 Report
The paper has been improved and the answer in Auther's comments addressed my questions.
Author Response
Thank you for your comments.
Reviewer 3 Report
1. Line 366, “….our previously established Macroscore system to this validation group.” requires citation.
2. Kindly incorporate the results of gene expression studies in the manuscript, and highlight the section of the discussion section that discusses the results of the gene expressions.
It is satisfactory.
Round 4
Reviewer 3 Report
1. The authors have used a scoring system that has no experimental validation. Keeping in consideration that the methodology is based on clinical samples, we expect to be conservative in our methodology, and apply only those methodology that has been previously established. Alternatively, we may experimentally establish a new scoring system that would require corroboration from existing system. Based on this particular lacuna of using a scoring system that has not been validated, we cannot concur with the conclusion has claimed by the authors. Hence, we regret to reject the manuscript.